# Lack of Phylogenetic Differences in Ectomycorrhizal Fungi among Distinct Mediterranean Pine Forest Habitats

**DOI:** 10.3390/jof7100793

**Published:** 2021-09-24

**Authors:** Irene Adamo, Carles Castaño, José Antonio Bonet, Carlos Colinas, Juan Martínez de Aragón, Josu G. Alday

**Affiliations:** 1Joint Research Unit CTFC-AGROTECNIO-CERCA, Av. Alcalde Rovira Roure 191, E25198 Lleida, Spain; jantonio.bonet@exchange.ctfc.es (J.A.B.); mtzda@ctfc.es (J.M.d.A.); josucham@gmail.com (J.G.A.); 2Department of Crop and Forest Sciences, University of Lleida, Av. Alcalde Rovira Roure 191, E25198 Lleida, Spain; carlos.colinas@udl.cat; 3Department of Forest Mycology and Plant Pathology, Swedish University of Agricultural Sciences, SE-75007 Uppsala, Sweden; carles.castanyo@c.ctfc.cat; 4Forest Science and Technology Centre of Catalonia, Ctra. Sant Llorenç de Morunys km 2, E25280 Solsona, Spain

**Keywords:** DNA metabarcoding, phylogenetic structure, habitat filtering

## Abstract

Understanding whether the occurrences of ectomycorrhizal species in a given tree host are phylogenetically determined can help in assessing different conservational needs for each fungal species. In this study, we characterized ectomycorrhizal phylogenetic composition and phylogenetic structure in 42 plots with five different Mediterranean pine forests: i.e., pure forests dominated by *P. nigra*, *P. halepensis*, and *P. sylvestris*, and mixed forests of *P. nigra-P. halepensis* and *P. nigra-P. sylvestris*, and tested whether the phylogenetic structure of ectomycorrhizal communities differs among these. We found that ectomycorrhizal communities were not different among pine tree hosts neither in phylogenetic composition nor in structure and phylogenetic diversity. Moreover, we detected a weak abiotic filtering effect (4%), with pH being the only significant variable influencing the phylogenetic ectomycorrhizal community, while the phylogenetic structure was slightly influenced by the shared effect of stand structure, soil, and geographic distance. However, the phylogenetic community similarity increased at lower pH values, supporting that fewer, closely related species were found at lower pH values. Also, no phylogenetic signal was detected among exploration types, although short and contact were the most abundant types in these forest ecosystems. Our results demonstrate that pH but not tree host, acts as a strong abiotic filter on ectomycorrhizal phylogenetic communities in Mediterranean pine forests at a local scale. Finally, our study shed light on dominant ectomycorrhizal foraging strategies in drought-prone ecosystems such as Mediterranean forests.

## 1. Introduction

Ectomycorrhizal fungi are essential organisms in forests, as they form symbiotic relations with trees providing them nutrients in exchange for photosynthetic carbon [1,2,3]. Some ectomycorrhizal fungi are host specific [3,4,5,6] and are influenced by tree species as well as by soil abiotic factors such as pH and nutrient availability [7,8,9,10]. Therefore, host effect and abiotic soil parameters are often fundamental drivers of ectomycorrhizal community assembly [11,12,13,14,15,16]. Moreover, previous studies showed that ectomycorrhizal taxonomic community composition does not significantly change between Mediterranean congeneric pine species [17]. Nevertheless, how ectomycorrhizal fungi are phylogenetically structured among Mediterranean pine host species and whether at both taxonomic and phylogenetic level respond to similar abiotic factors has not been assessed yet. Previous studies showed that ectomycorrhizal responses to climate warming are modulated by host plant performance and nutrient availability [18,19,20]. Therefore, it is crucial to disentangle whether these drivers influence ectomycorrhizal phylogenetic composition and structure, to better understand forest ecosystem functioning [21].

Phylogenetic analyses are useful tools to estimate the relative importance of evolutionary and ecological forces structuring communities [12,22,23]. In this regard, phylogenetic indices have been implemented to calculate the phylogenetic relatedness of an observed community and compare the value to expectations of community assembly under neutral processes from a regional species pool [24]. Therefore, these indices enable us to characterize whether communities are more phylogenetically related (phylogenetic clustering) or less phylogenetically related (phylogenetic overdispersion) than expected by chance [22,23,24]. In general, habitat filtering is the dominant assembly process when closely related species that share similar traits are selected to coexist within the community (i.e., phylogenetic clustering). In contrast, competition processes occur when distantly related species with dissimilar traits are selected to co-occur within a community (i.e., phylogenetic overdispersion [22], while the random phylogenetic structure is detected when none of the above processes are inferred [22,23,24,25]. For example [26], observed phylogenetic clustering of Agaricomycotina communities (including mycorrhizal and saprotrophs) and observed that xeric oak-dominated forests acted as a filter for these communities. Likewise [27], found phylogenetically clustered arbuscular mycorrhizal communities along an altitudinal gradient and observed that environment was the primary ecological factor structuring these communities, either via changes in host plant or fungal niches. Although the ecological processes filtering communities have recently received criticism [28], investigating the communities’ phylogenetic responses to the environment in different ecosystems is fundamental to understand the mechanisms that structure communities [29,30]. However, how ectomycorrhizal communities are phylogenetic structured in Mediterranean pine forests has not been studied yet.

The description of phylogenetic relations between ectomycorrhizal fungi might help to understand the evolutionary ecology of traits, species, and entire communities [31]. In this regard, exploration types of ectomycorrhizal fungi represent an important group of functional traits, which are defined according to the hyphal morphology, i.e., long distance, medium distance, medium distance fringe, short distance, or contact exploration types [32]. The hyphal morphology determines access to distinct nutrient sources, for example, nitrogen (N) [33,34,35]. Ectomycorrhizal species with short, contact, and medium smooth distance exploration types may preferentially use soluble inorganic forms of N close to the host roots due to the lack the enzymes to access organic N forms [33,36]. Conversely, some fungi have enzymes (i.e., fenton peroxidase) to access insoluble N substrates such as organic substrates and they usually show medium mat and long-distance exploration types [35,36,37]. However, long and medium fringe exploration types might demand higher carbon cost on the host than shorter distance exploration types [18,32], therefore species with shorter exploration types may be favored under stressful conditions [18,38]. In this regard, several studies have addressed ectomycorrhizal exploration types’ responses to environmental drivers [39,40,41,42], however, the phylogenetic pattern of the trait in Mediterranean ecosystems has rarely been assessed. Thus, understanding the phylogenetic relationships between ectomycorrhizal species and the evolution of hyphal morphologies in the current climate change context might shed light on the future impacts on Mediterranean ecosystem functioning.

In this study, we aim to characterize the ectomycorrhizal phylogenetic composition and phylogenetic structure in 42 plots of five different Mediterranean pine forests: i.e., pure forests dominated by *P. nigra*, *P. halepensis*, and *P. sylvestris*, and mixed forests of *P. nigra-P. halepensis* and *P. nigra-P. sylvestris*. In line with the above premises, we hypothesized that:Considering that *P. halepensis*, *P. nigra*, and *P. sylvestris* are phylogenetically closely related [43,44,45], we expect that ectomycorrhizal phylogenetic composition, structure, and diversity will not be different among them due to co-evolutionary processes [46]. Previous studies have identified that ectomycorrhizal taxonomic composition is influenced by soil parameters followed by geographical distance [27,47] Thus, we hypothesized that soil physico-chemistry will act as the main habitat filter on ectomycorrhizal phylogenetic composition [48]. Finally, among abiotic filters pH, P, and CN ratio strongly influenced ectomycorrhizal taxonomic community composition [17]. Here, we tested if these filters would act similarly over ectomycorrhizal phylogenetic composition. In Mediterranean ecosystems, soil N might not be limited due to warmer temperatures which may enhance N mineralization by increasing decomposition of the organic matter [17,49]. Therefore, short exploration types could uptake nutrients close to the host roots. Here, we expected that short and contact exploration types will be dominant, thus, both traits will be overrepresented and dispersed across the ectomycorrhizal phylogenetic tree in comparison with medium and long-term exploration types. 

## 2. Materials and Methods

### 2.1. Sites Selection

The study was conducted in the mountainous pre-Pyrenees region of Catalonia, North-eastern Spain (Appendix A) in a set of long-term monitoring plots in which fungal fruiting has been recorded for ~20 years [50]. The region is under the influence of the Mediterranean climate, with a summer drought period from June to August and mean annual temperatures from 6 to 9 °C with most of the precipitation occurring in spring and autumn [51]. The 42 pine forests were randomly selected from the 579 sites included in the Forest Ecological Inventory of Catalonia carried out by Centre de Recerca Ecològica i Aplicacions Forestals (CREAF, Barcelona, Spain, 1992), trying to preserve even-aged forest. From the total 42 forests, 32 correspond to pure pine forests, with 14 plots corresponding to *P. nigra* and *P. sylvestris* species and 4 to *P. halepensis*, whilst 10 plots were mixed plots (7 mixed plots of *P. sylvestris* and *P. nigra* species and 3 plots dominated by *P. nigra* and *P. halepensis*). The main features of the study plots are summarized in Table 1 and Appendix A.

### 2.2. Soil Sampling

Soils were sampled during the Autumn season (October and November) in 2009. In each of the selected forest stands, a 10 m × 10 m plot was established in the center for long-term monitoring of fungal fruiting [51]. In each plot, we took four soil subsamples, i.e., one per plot-side [52], with a rectangular steel drill (a 30 cm depth and a 6 × 4.5 cm width). The four soil subsamples were pooled in the field and around 1 kg of the mixed sample was placed on ice and taken to the laboratory for fungal DNA extraction. A similar procedure was followed for soil samples to determine soil physico-chemical parameters.

### 2.3. Soil Analysis

Soil samples were analyzed using the methodology described in [53]. Each sample was air-dried, sieved (≤2 mm mesh), and soil texture (clay, sand, and lime proportions) was analyzed using the Bouyoucos—method [54]. Soil pH and electrical conductivity (EC) using a conductivity meter in a 1:2.5 soil:deionized water slurry [55]. Total nitrogen concentration using the Kjeldahl method [56]. Moreover, available phosphorus concentration using the Olsen method [57]; total organic matter and total carbon concentration using the Walkley-Black method [58]. Finally, exchangeable cations such as sodium (Na), potassium (K+) and magnesium (Mg2+) with atomic absorption spectroscopy after extraction with 1 N ammonium acetate (pH 7; [55,56,57,58,59]).

### 2.4. Fungal Community and Bioinformatic Analysis

Fungal DNA was extracted from 0.5 g of homogenized soil using the NucleoSpin^®^ NSP soil kit (Macherey-Nagel, Duren, Germany) following the manufacturer’s protocol. Fungal internal transcribed spacer 2 (ITS2) region was amplified in a 2720 Thermal Cycler (Life Technologies, Carlsbad, CA, USA) using the primers gITS7 [60], ITS4, and ITS4A [61,62]. We optimized the number of PCR cycles in each sample aiming for weak to medium PCR bands at the agarose gels, which was achieved in most of the samples by using 21–26 cycles. The final concentrations in the PCR reactions, PCR conditions, DNA purification and sequencing, and bioinformatics analyses were as explained by Adamo et al. (2021). Sequence data are archived at NCBI’s Sequence Read Archive under accession number PRJNA641823 (www.ncbi.nlm.nih.gov/sra, accessed on 25 June 2020).

### 2.5. Taxonomic and Functional Identification

We taxonomically identified the 600 most abundant OTUs, which represented 93% of the total sequences. We selected the most abundant sequence from each OTU for taxonomic identification, using PROTAX software [63] implemented in PlutoF, using a 50% probability of correct classification (called “plausible identifications”) [63]. These identifications were confirmed and some of them improved using massBLASTer in PlutoF against the UNITE [64]. Taxonomic identities at species level were assigned based on >98.5% similarity with database references, or to other lower levels using the next criteria: genus on >97%, family on >95%, order on >92%, and phylum on >90% similarity. OTUs were assigned to the following functional guilds: (a) root-associated basidiomycetes, (b) root-associated ascomycetes, (c) molds, (d) yeasts, (e) litter-associated basidiomycetes, (f) litter-associated ascomycetes, (g) pathogens, (h) moss-associated fungi, (i) soil saprotrophs (saprotrophic taxa commonly found in N-rich mineral soils), (j) unknown function, based on the UNITE database, DEEMY (www.deemy.de) or FUNGuild [65]. ECM species were assigned to exploration types according to the DEEMY database [66,67].

### 2.6. Phylogenetic and Statistical Analyses

The ghost-tree approach [68], which allows sequence data to be integrated into a single tree, was used to reconstruct the fungal phylogenetic tree. Foundation phylogeny at the family level was derived by (Treebase ID S20837) [69], following methodology and was based on the sequences of six genes 18sS rRNA, 28S rRNA, 5.8S rRNA, translation elongation factor 1-α (tef1α), and RNA polymerase II (two subunits: RPB1 and RPB2) [70]. 

Statistical analyses were implemented in the R software environment (version 3.6.1, R Development Core Team 2019). The *ape* package was used to load and manipulate the phylogenetic tree in newick format [71], while the *phyloseq* package was used to import and handle OTU counts [72], taxonomic assignments, and associated phylogenetic tree. The *philR* package was used to analyze compositional data using the phylogenetic tree information [73]. The *picante* package was used to calculate the ectomycorrhizal phylogenetic structure indices (NRI and NTI) and Faith’s phylogenetic diversity [74,75]. The *vegan* package was used for the multivariate analyses [76].

For all compositional analyses, the ectomycorrhizal species abundance matrix was previously filtered to exclude the taxa that were not seen in at least 10% of samples to eliminate random noise. We analyzed ectomycorrhizal phylogenetic community composition using philR which enables us to transform compositional data into an orthogonal unconstrained space with phylogenetic and evolutionary interpretation [73]. First, PhilR Isometric Log Ratio transformations were built from the phylogenetic tree utilizing a weighted reference [73], then a Euclidean distance matrix was built from the philR transformed data. After that, Redundancy analysis, (RDA function “*rda*”) was used to visualize ectomycorrhizal phylogenetic compositional differences between tree host species. Moreover, ectomycorrhizal phylogenetic differences between tree host species were tested using permutational multivariate analyses of variance (PMAV, function “*adonis*”) on the Euclidean dissimilarity matrix based on the philR transformed data. To test the phylogenetic structure of ectomycorrhizal communities between tree host species were calculated the standardized effect size of mean pairwise distances and mean nearest taxon distances using *ses.mpd* (Standardized effect of mean pairwise distances in communities) and *ses.mntd* (Standardized effect of nearest taxon index in communities) functions from *picante*. In each stand type, we compared the MPD and MNTD values with the MPD and MNTD distributions of random communities in order to identify whether communities were more over-dispersed or under-dispersed than expected by chance. We used the *independentswap* null model, which randomizes community data matrix with the independent swap algorithm maintaining species occurrence frequency and sample species richness, to construct from 9999 randomly assembled communities [77]. After calculating SES.MPD and SES.MNTD, the values were multiplied by −1 as these values are equivalent to −1 times NRI (net relatedness index) and NTI (nearest taxon index), respectively. Importantly, an increase in the NRI value indicates increasing phylogenetic clustering (or decreasing overall relatedness) of a set of species relative to the source pool [25]. On the other hand, the nearest taxon index (NTI) is a standardized measure of the mean phylogenetic distance to the nearest taxon in each sample/community [25]. The NRI measures the standardized effect size of the mean phylogenetic distance (MPD), which estimates the average phylogenetic relatedness between all possible pairs of taxa in a community. The NTI calculates the mean nearest phylogenetic neighbor among the individuals in a community. The ectomycorrhizal phylogenetic diversity comparisons between tree host species were done using Faith’s PD phylogenetic diversity index with the function *pd*. Moreover, to assess the phylogenetic relationships among species change across space, we computed multiple-site phylogenetic turnover, nestedness, and phylo-beta diversity (Sorensen similarity index) per tree host species using “*phylo.beta.multi*” function in the betapart package [78].

Second, variation partitioning (function “*varpart*”) was used to test the relative importance as variation sources of geographical distances, soil parameters, and stand structure in ectomycorrhizal phylogenetic composition (philR transformed data) and structure (NRI, NTI). To avoid multicollinearity before variation partitioning analysis highly correlated environmental variables were removed (*r* > 0.7). The geographical distances included were previously evaluated using principal coordinates of neighbors’ matrices spatial eigenvectors (PCNM, *pcnm* function) based on UTM coordinates of the sampled stands with Euclidean distances. Thus, significant spatial eigenvectors were forward selected and the selected spatial eigenvectors were used as explanatory variables in the variation partitioning, together with soil (Sand content, K, Mg, organic matter, Na, N, P, water pH, and CN ratio) and stand structural variables (Tree species, Altitude, Slope, Trees per hectare, and Basal Area). The significance of each partition was tested using multivariate ANOVAs. Moreover, to evaluate the effect of pH, CN, and P on the ectomycorrhizal phylogenetic composition we conducted a redundancy analysis (*rda* function). In addition, the *sm.density.compare* (*n* of permutations = 999) from the *sm* package was used to randomly assign pH values between the five tree hosts and estimate how different the densities were using a permutational test of density equality [79]. Lastly, to visualize if the ectomycorrhizal phylogenetic communities were clustering across the pH gradient, we performed a hierarchical cluster analysis on the ectomycorrhizal phylogenetic compositional data based on the Euclidian distance matrix using the function *hclust* in the *stats* package. 

Finally, a binary data matrix was compiled with ectomycorrhizal exploration traits (contact, short, medium smooth, medium mat, medium-fringe, and long). Then, we calculated the trait *ses.mpd* and *ses.mntd* using the *independentswap* null model to assess trait structure following the same methodology for communities. Finally, to test for a phylogenetic signal to exploration types, K’ Blomberg statistics were calculated for the presence of the traits using the function *MultiPhylosignal* in the *picante* package [74,80]. Moreover, the traits were visualized on the phylogenetic tree by plotting the exploration types at the tips of the phylogenetic tree following [74].

## 3. Results

### 3.1. Ectomycorrhizal Phylogenetic Description

The hybrid phylogenetic tree of ectomycorrhizal fungi was consistent with Mikryukov et al. (2020) (Figure 1). The families Sebacinaceae, Clavulinaceae, and Hydnaceae clearly formed a monophyletic group, while Bankeraceae Thelephoraceae, Russulaceae, and Albatrellaceae formed two distinct clades (Figure 1). Moreover, two other family groups were identified, one including Atheliaceae, Sclerodermataceae, Boletaceae, Gomphidiaceae, and Suillaceae, and the other including Tricholomataceae, Amanitaceae, Hydnangiaceae, Cortinariaceae, Hymenogastraceae, and Inocybaceae (Figure 1). Finally, the most abundant species in each tree host were indicated in Appendix A. 

### 3.2. Ectomycorrhizal Phylogenetic Composition, Structure, and Diversity

There were no significant differences in the ectomycorrhizal phylogenetic composition among tree host species (*r*^2^ = 0.10, F_(4,41)_ = 1.12, *p* = 0.281). The RDA and the sd-ellipses based on the philR Euclidean distance matrix clearly showed that all forest types were overlapping at the ordination center (Figure 2). Redundancy analyses resulted in two main axes that explained together 22% of the variance. However, *P. halepensis-nigra*, *P. halepensis*, and *P. nigra* communities were less spread (homogeneous), while, *P. sylvestris* and *P. sylvestris-nigra* communities were more overdispersed in the ordination space (heterogeneous). Regarding ectomycorrhizal phylogenetic structure, no significant difference was detected for NRI (F_(4,41)_ = 0.26, *p* = 0.901) between tree host species. Positive mean values of NRI were detected in *P. halepensis* (0.51 ± 0.09), indicating ectomycorrhizal higher phylogenetic clustering. *P. nigra-halepensis* (0.14 ± 0.83), *P. sylvestris-nigra* (0.06 ± 0.25) and *P. sylvestris* (0.05 ± 0.27), and *P. nigra* (−0.02 ± 0.26) showed dispersion of NRI values positive and negative around 0 (Figure 3a). However, we detected significant differences in NTI values (F_(4,41)_ = 2.96, *p* = 0.031) between tree host species. Mean positive NTI values were detected across all tree host species, except in *P. sylvestris-nigra* (−0.46 ± 0.37), indicating ectomycorrhizal phylogenetic clustering in *P. halepensis*, *P. nigra-halepenesis*, while *P. nigra*, *P. sylvestris* were not clearly defined, with values around 0, and a marginal phylogenetic overdispersion was detected in *P. sylvestris-nigra* (Figure 3b).

The ectomycorrhizal phylogenetic diversity analysis showed no significant differences between tree host species (F_(4,41)_ = 0.92, *p* = 0.458, Figure 3c), with PD mean values ranging from 6.6 of *P. nigra* and 8.3 *P. sylvestris* (Figure 3c). In addition, analysis of multiple-site phylogenetic similarities showed that total beta diversity values were similar across host tree species (Appendix A), although, species turnover resulted strongly higher than species nestedness across the tree host species and with similar values, except for *P. nigra-halepensis* (Phylo beta.sim: 0.40; Phylo beta.sne: 0.14; Appendix A).

### 3.3. Main Drivers of Ectomycorrhizal Phylogenetic Composition and Structure

When testing the relative importance of geographic distance, soil parameters, and stand structure on ectomycorrhizal phylogenetic composition, soil accounted for the greatest proportion of the total variance (4%) followed by geographic distance, however, these fractions were not significant (*p* > 0.05, Figure 4a). Moreover, stand structure, soil, and geographic distance shared 4% of the total variance. Conversely, when the phylogenetic structure was analyzed, stand structure, soil, and geographical distance shared an 8% proportion of variation, while stand structure accounted for 4% of the total variance (*p* > 0.05) (Figure 4b). Finally, the phylogenetic structure was marginally influenced by soil (2%) and not by geographic distance.

pH was the only significant soil predictor influencing phylogenetic composition (Variance = 0.55, F = 2.76, *p* = 0.002). Thus, the distribution of pH values was significantly different across tree hosts (*p* = 0.041). Moreover, when pH densities were compared only *P. sylvestris* and *P. nigra* differed significantly (*p* = 0.009) showing a larger left tail towards lower pH values (Appendix A). The hierarchical clustering of the ectomycorrhizal phylogenetic composition showed that communities were clustered into two main groups (Appendix A), Here, the group composed of *P. halepensis*, *P. sylvestris*, and *P. sylvestris-nigra* communities clustered at lower pH values (<7), while *P. nigra* and *P. nigra-halepensis* communities only occurred at higher pH values (>7) (Appendix A).

### 3.4. Trait Evolution of the Exploration Types

When the exploration traits were visualized on the phylogenetic tree, 59 OTUs out of 184 had short exploration types, up to 53 had contact exploration types, while 39 OTUs and 25 OTUs had medium fringe and medium smooth exploration types. Conversely, medium mat and long exploration types were the least abundant with 9 and 8 OTUs, respectively. Finally, we did not find any phylogenetic signal for any exploration type (0.25 < K < 0.77, *p* > 0.05), as exploration types were dispersed across the phylogenetic tree (Figure 5).

## 4. Discussion

The results of our phylogenetic study on ectomycorrhizal communities in Mediterranean pine forests showed that phylogenetic composition, structure, and diversity were similar among habitats with distinct pine tree hosts. However, significant differences were found in nearest taxon index values between *P. nigra-halepensis* and *P. sylvestris-nigra*, probably not directly caused by differences in tree hosts but due to higher differences in the local abiotic conditions in *P. sylvestris-nigra* than in *P. halepensis-nigra* sites. Moreover, we detected a weak abiotic filtering effect on the ectomycorrhizal phylogenetic compositional variation, being pH the only variable among soil variables that significantly influence the ectomycorrhizal phylogenetic community. This finding suggests that pH acts as a strong abiotic filter on the ectomycorrhizal community at both phylogenetic and taxonomic levels [17]. In contrast, ectomycorrhizal phylogenetic structure variation was marginally influenced only by the shared effect of stand structure, soil, and geographic distance. Therefore, the phylogenetic structure may be indirectly influenced by other processes (i.e., competition; [30]) not directly tested in this study. Finally, we identified that short and contact exploration types were the most abundant in these forest ecosystems. Conversely, long exploration types were the least abundant, although there was no phylogenetic signal since exploration types were dispersed across the phylogenetic tree.

### 4.1. Ectomycorrhizal Phylogenetic Description

Our study allowed us to investigate the phylogenetic relationships between 256 OTUs using a multiple gene tree at family level as a foundation tree which allows us to build a better-supported tree (Figure 1a) [70]. Also, we were able to identify monophyletic groups of families, such as Sebacinaceae, Clavulinaceae, and Hydnaceae, and Atheliaceae, Sclerodermataceae, Boletaceae, Gomphidiaceae, and Suillaceae, however, this last clade formed a paraphyletic group with Russulaceae and Albatrellaceae. Moreover, two other family groups were identified, one including Bankeraceae, Thelephoraceae, Sclerodermataceae, Boletaceae, Gomphidiaceae, and Suillaceae, and the other including Tricholomataceae, Amanitaceae, Hydnangiaceae, Cortinariaceae, Hymenogastraceae, and Inocybaceae. Therefore, the resolved phylogenetic tree resulted in a strong backbone for the downstream analyses as the level of resolution allows us to perform reliable phylogenetic diversity analyses [81]. Finally, disentangling the ectomycorrhizal phylogenetic community structure in our study region, where the current climate change may lead to changes in ecosystems functioning, is crucial to predict the impacts on ectomycorrhizal taxonomic and phylogenetic community composition and diversity [82].

### 4.2. Ectomycorrhizal Phylogenetic Composition, Structure, and Diversity

Our results demonstrated that ectomycorrhizal phylogenetic community and diversity were not significantly different among pine tree host species or in NRI values, although there were differences in NTI values between *P. sylvestris-nigra* and *P. nigra-halepensis*. These results are in accordance with previous taxonomical studies on ectomycorrhizal communities between congeneric tree hosts [9,83], in which a lack of phylogenetic differences was observed. Similarly, ectomycorrhizal community composition was not different between phylogenetically related pines in China [84]. In contrast, several studies reported taxonomical differences between ectomycorrhizal communities between hosts of different families or genera [7,85]. Thus, it seems that at both taxonomic and phylogenetic levels, ectomycorrhizal communities are not varying significantly among phylogenetically close related tree hosts [14].

Similar phylogenetic studies detected phylogenetic clustering of Agaromycotina communities in xeric oak-dominated forests and concluded that oak acted as the main habitat filter [26]. Here, our results showed an opposite trend, with no significant differences in ectomycorrhizal phylogenetic structure and diversity between pine tree hosts. However, we observed significant differences in phylogenetic dispersion among habitats with distinct pines hosts. For example, ectomycorrhizal species in *P. halepensis-nigra* forest resulted in phylogenetic clusters, while in *P. sylvestris-nigra* were slightly more overdispersed at the tip of the phylogeny (Figure 3a), probably due to the low number of *P. nigra-halepensis* sites which may have caused underestimation of the differences between phylogenetic taxa. In this regard, the three *P. nigra-halepensis* sites showed similar soil properties (i.e., values range from pH: 8.18–8.38, CN: 11–151, P: 2–5, N: 0.12–0.16), which may have resulted in the occurrence of closely related species that are adapted to these similar abiotic conditions. These results may imply that Mediterranean pine host tree species are weak habitat filters for ectomycorrhizal fungi, probably due to a lack of host specificity among congeneric hosts. Thus, our results are in agreement with the hypothesis that the lack of phylogenetic composition, structure, and diversity between pine host species may be partially explained by possible conserved symbiosis between *Pinus* and ectomycorrhizal fungi [86].

Finally, we found high and similar turnover values in all the tree host species forest, while nestedness was significantly lower, except in the case of *P. nigra-halepensis* forest. It seems that both environmental filtering by soil and dispersal limitation may, to a certain extent, promote species replacement among sites [87]. However, in *P. nigra-halepensis* forest higher nestedness might indicate local species loss probably due to its soil site conditions that resulted in the occurrence of a locally adapted subset of species.

### 4.3. Main Drivers of Ectomycorrhizal Phylogenetic Composition and Structure

In this study, we observed that soil parameters influenced ectomycorrhizal phylogenetic composition, while phylogenetic structure variation was primarily influenced by the shared effect of the three environmental filters. However, these three fractions were not significant and explained a residual amount of variation, thus, the second hypothesis is not accepted. Although previous studies have identified that soil parameters are the main drivers of taxonomic ectomycorrhizal community variation in Mediterranean pine forest [17], here, soil parameters were marginally important in driving ectomycorrhizal phylogenetic composition. This may imply that at the phylogenetic level, the lack of strong abiotic gradients results in the occurrence of non-closely related species which are adapted to heterogeneous but not specific environmental conditions [88].

Soil properties have been widely described as a strong abiotic filter on taxonomic fungal communities at different spatial scales [14,62,89,90]. In contrast, we observed a weak abiotic filtering effect of soil physico-chemistry on phylogenetic community composition, with pH resulting in the only significant variable. The importance of pH as an influential variable over ectomycorrhizal community composition at local and regional scales has been widely described [87,90,91]. However, in view of our results, it seems that pH acts as an abiotic filter at both taxonomic and phylogenetic levels [87,92,93]. In addition, our results showed that a left tail of *P. sylvestris* and *P. sylvestris-nigra* in the distribution of the pH values resulting in a wider niche for species adapted to low pH values (Appendix A). In this regard, phylogenetic fungal communities were more clustered at lower values of pH (<7), thus, it seems that lower pH values might result in the occurrence of only adapted fungal species that can grow and maintain cellular function in acidic environments [94], causing higher phylogenetic similarity [87,95,96].

Regarding phylogenetic structure, stand structure alone explained a proportion of its variation although this was not significant. However, stand structure, soil physico-chemistry, and geographic distance shared an important proportion of variance. In Mediterranean ecosystems, the influence of stand structural variables on fungi has already been assessed over mushroom yields with important effects [81]. Although weakly, differences in stand structural variables may result in the occurrence of different less phylogenetically related fungal species that are better adapted to certain local forest conditions. In view of our results, we argue that ectomycorrhizal phylogenetic structure is more importantly influenced by the combined effect of all environmental variables. Hence, phylogenetic relatedness between species decreases with increasing geographic distance, differences in stand structure, and soil conditions. Finally, we hypothesize that there may be other processes influencing the ectomycorrhizal phylogenetic community, such as competition for space and resources [30]. that were not directly tested, therefore further studies are needed to further disentangle whether other processes influence phylogenetic structure in these ecosystems.

### 4.4. Trait Evolution of the Exploration Types

We observed that 51% of the ectomycorrhizal species had short and contact exploration types, 21% and 13% of the species had medium fringe and medium exploration types, respectively, and only 4% of the species had long exploration types. Similarly, [38] found that in P. pinaster-dominated Mediterranean forests, long distance were the least abundant exploration types, while short and contact types were dominating the community. Moreover, our results showed that traits were dispersed across the phylogenetic tree (Figure 5). Thus, the third hypothesis is accepted. In this regard, the dispersion of traits across the phylogenetic tree suggests that even more distant related species showed the same exploration type, resulting in a random trait pattern with a lack of phylogenetic signal. In addition, [38] found that mycorrhizal species with long distance exploration types were less abundant under drier conditions, whereas short-distance and contact type species increased. Recent studies suggest that drier conditions may favor short-contact types [18,38]. Similarly, based on our results, we argue that dispersion of short and contact exploration types might be an adaptation to the Mediterranean stress conditions where the limiting factor is water and not nutrients. Therefore, having medium mat and long exploration types might be a disadvantage due to their higher C demand on the host [18,32,38]. At the same time, in northern and temperate ecosystems, soil N is a limiting nutrient [81], and previous studies have shown that species with long medium mat exploration occurs in soils where N is limiting and patchily distributed [37,42], while short exploration types are more efficient in up-taking soluble inorganic N [36]. However, despite these observed trends and since exploration types of mycorrhizae represent a distinct set of fungal traits, the use of exploration types to study fungal trait responses to environmental changes can be often misleading, and further research should be addressed. In any case, previous work in this area showed a lack of N effect on mycorrhizal communities in Mediterranean pine forests [17], therefore, as N is not limiting it can be easily captured by ectomycorrhizal fungi close to the host with no need of investing in high biomass exploration types.

Finally, we acknowledge the accuracy of the ITS2 region in species identification and resolution [97], but also its limitation in phylogenetic applications due to its high variability [70,98]. However, the use of a backbone phylogenetic tree at family level constructed from multiple gene sequences provides a sufficient taxonomic resolution, thus can be an accurate predictor of phylogenetic diversity metrics [99]. Moreover, it is known that the identification of basidiomycetes through ITS2 amplification is more efficient than in other taxa (i.e., Ascomycetes) [100]. Therefore, future studies aiming to disentangle fungal phylogenetic patterns in community structure should include a robust backbone phylogenetic tree and at least the whole ITS region.

## 5. Conclusions

In this study, we found no differences neither in ectomycorrhizal phylogenetic community composition nor structure and diversity, indicating that ectomycorrhizal communities at both phylogenetic and taxonomic levels do not change among phylogenetically closely related tree hosts. Moreover, soil parameters only had a marginal filtering effect on ectomycorrhizal phylogenetic variation as pH resulted in the only significant driver of the phylogenetic community. In this regard, our results showed that pH acts as the broadest abiotic filter of ectomycorrhizal communities at a local scale.

Conversely, the ectomycorrhizal phylogenetic structure was marginally influenced by the combined effect of soil, stand structure, and geographic distance, indicating that phylogenetic structure is mainly influenced by their combined effect.

Finally, short and contact distance were the dominant exploration types, as they may be favored under drought stress conditions but also under high nutrient availability. Our results shed light on the drivers of ectomycorrhizal phylogenetic community variation in Mediterranean pine forests, being fundamental to get a better insight on the drivers of community assembly and ecosystem functioning. Nevertheless, further research on ectomycorrhizal phylogenetic communities is needed to better understand how changes in deterministic processes will affect ectomycorrhizal communities and forest ecosystems’ functioning.

## Figures and Tables

**Figure 1 jof-07-00793-f001:**
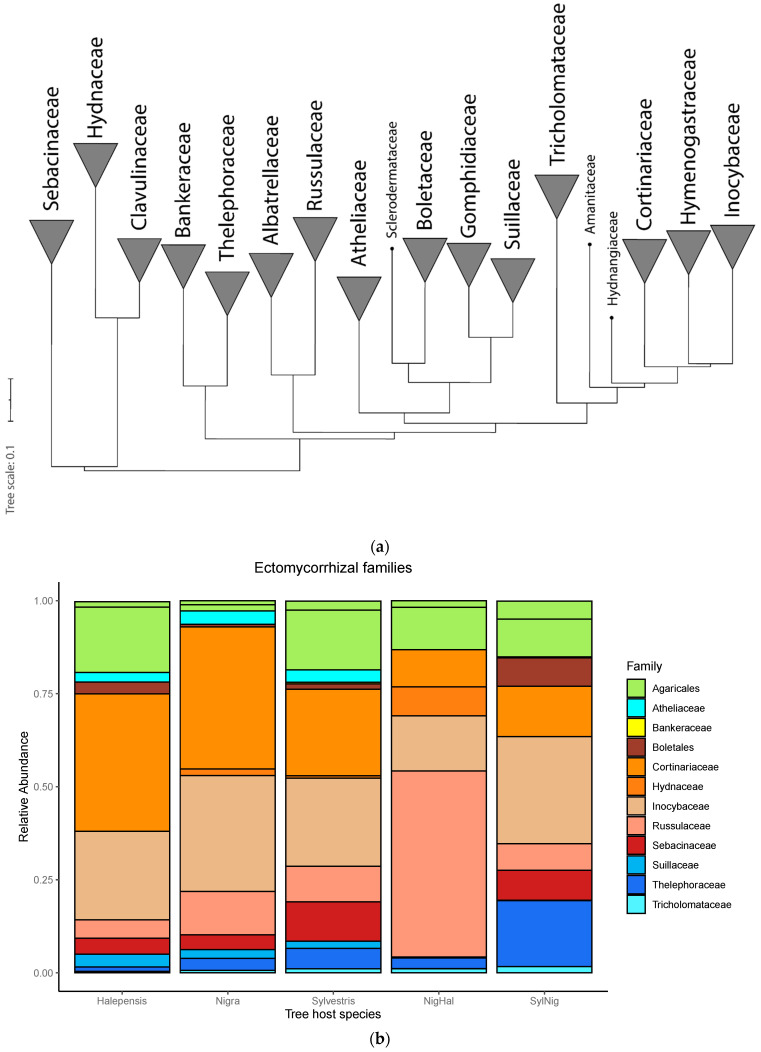
(**a**) The hybrid phylogenetic tree of ectomycorrhizal families based on the foundation phylogeny derived by Zhao et al. (2017), based on the sequences of six genes 18sS rRNA, 28S rRNA, 5.8S rRNA, translation elongation factor 1-α (tef1α) and RNA polymerase II. (**b**) Relative abundance of the most abundant ectomycorrhizal families.

**Figure 2 jof-07-00793-f002:**
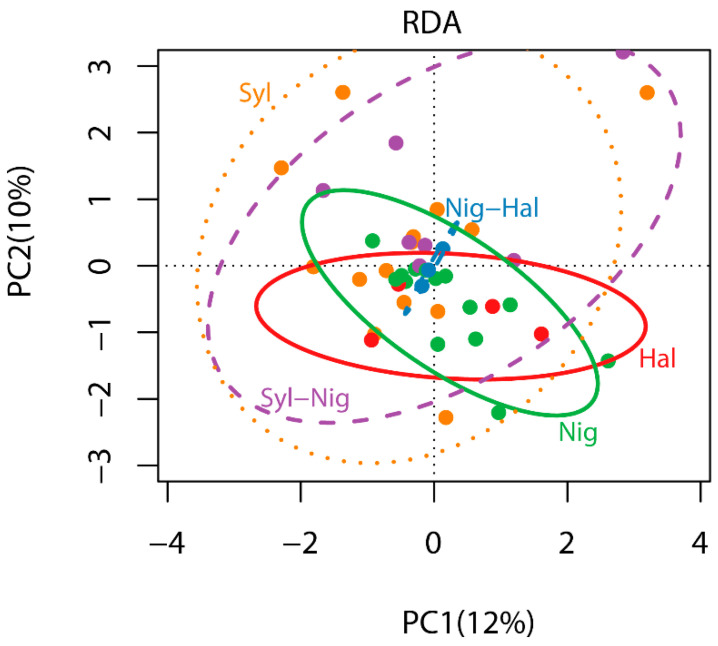
philR RDA ordination based on Euclidean distance matrix displaying ectomycorrhizal phylogenetic community composition of *P. halepensis*, *P. nigra-halepensis*, *P. nigra*, *P. sylvestris-nigra* and *P. sylvestris* forest and the sd ellipses of each forest.

**Figure 3 jof-07-00793-f003:**
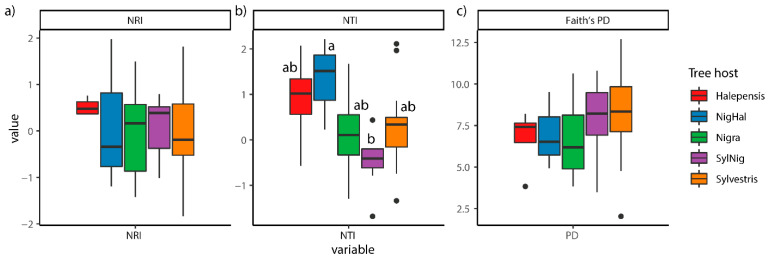
Boxplots displaying (**a**) Net Relatedness Index (NRI) values (**b**) Nearest Taxon Index (NTI) (**c**) Faith’s PD values between tree host species (Halepensis: *P. halepensis*, NigHal: *P. nigra-halepensis*, Nigra: *P. nigra*, SylNig: *P. sylvestris-nigra* and Sylvestris: *P. sylvestris*). Means were compared using ANOVA and Tukey’s HSD tests, with letters denoting significant differences between host species.

**Figure 4 jof-07-00793-f004:**
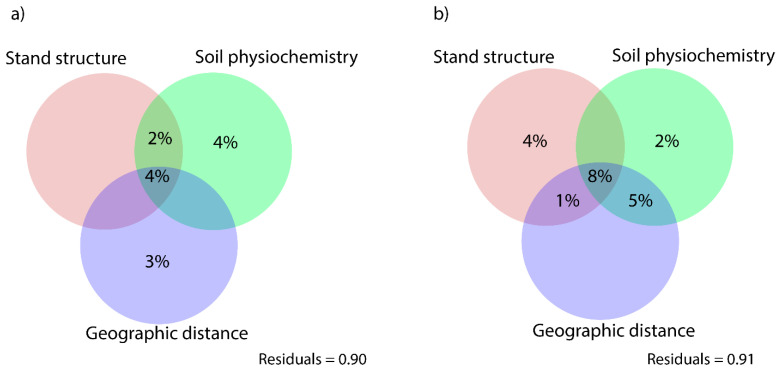
Variance partitioning analyses for (**a**) ectomycorrhizal phylogenetic composition and (**b**) ectomycorrhizal phylogenetic structure (NRI, NTI indices) in response to stand structure, soil physiochemistry and geographic distance. Values show the fraction of variation explained by each group of parameters, as well as the shared contribution of each combination of them.

**Figure 5 jof-07-00793-f005:**
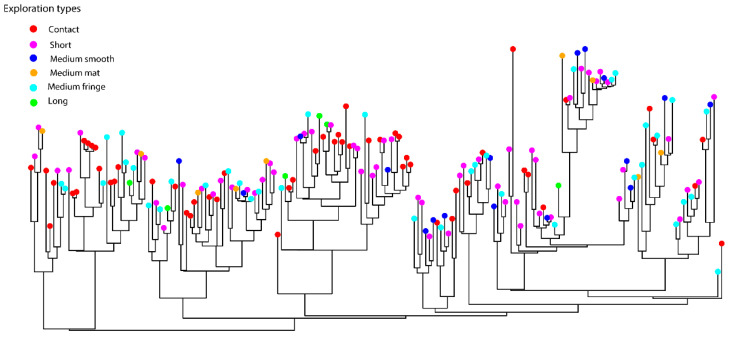
The hybrid phylogenetic tree displaying the distribution of the exploration types (Contact, short, medium smooth, medium fringe and long).

**Table 1 jof-07-00793-t001:** Table summarizing the main features of the study plots: BA (Basal area), Number of trees per hectare, Altitude, Slope, pH, CN ratio and P (Phosphorus). Ps: *P. sylvestris*, Pn: *P. nigra*, Ph: *P. halepensis*, Ps-Pn: *P. sylvestris-nigra*, Pn-Ph: *P. nigra-halepensis*.

Forest Type	Range	BA, m^2^ ha^−1^	N. of Tree Per Hectare	Altitude, m a.s.l	Slope, %	pH	CN Ratio	P
Ps	Min.	18.0	681	854	4	4.8	6.9	2
(14)	Mean	29.8	1362	1197	22	7.2	12.4	5.8
	Max.	41.5	1517	1615	37	8.3	19.5	9
Pn	Min.	16.1	638	397	5	8.0	4.0	3
(14)	Mean	27.7	1692	763	16	8.2	14.4	5.0
	Max.	39.1	2838	1040	32	8.4	21.3	9
Ph	Min.	24.0	1006	520	10	8.2	12.5	3
(4)	Mean	28.8	2093	612	16	8.3	13.6	4.8
	Max.	33.6	3088	661	34	8.4	14.8	6
Ps–Pn	Min.	11.5	477	1030	8	6.6	12.1	2
(7)	Mean	23.5	1161	1085	24	7.7	14.5	3.3
	Max.	31.8	2870	1148	31	8.3	19.8	5
Pn–Ph	Min.	17.6	1229	390	9	8.2	11.1	2
(3)	Mean	19.7	1806	469	12	8.3	13.2	4.0
	Max.	20.9	2761	577	13	8.4	15.4	5

## Data Availability

Available upon reasonable request.

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
