# Peer review of "Lack of Phylogenetic Differences in Ectomycorrhizal Fungi among Distinct Mediterranean Pine Forest Habitats"

_jof, 2021, doi:10.3390/jof7100793_

Round 1
Reviewer 1 Report
Overall, this piece of work did characterization of ectomycorrhizal fungi from five typical pine forests in Mediterranean area. This is interesting to general readers to understand their features and possible functions for the plant health under this climatic condition.
The major concern:
- EMF is usually strongly associated with plant roots. In this study, there is limited information about how their soil samples linked to their pine species host or root (seemed from top 30cm), it is difficult to make a convincing conclusion that tree host impacts were not significantly different.
- The author did several phylogenetic and statistical analyses, but the results did not comprehensively reflect the analyses that have been performed.
- There are 600 abundant OTUs, which could be classified into several functional groups, but I did not see any information about the relative abundance of each group.
- There is no detailed information about EMF profiles and characteristics from each pine forest.
- The author measured soil texture, but where is the data? The soil moisture and texture will have major impacts on fungi due to their impacts on water and oxygen availably.
- Some of the analyses are very confusing.
- For instance, the site description in Table 1, why using C:N ratio rather than TOC, TN, which are important to understand soil quality and fertility?
- Figure 1 phylogenetic tree, in addition to the major EMF cluster and their assigned families, it is also important to highlight whether their presence and/or relative abundance in individual forest types as it is one of the focus of this study.
- Figure 2 it is not common to use Euclidean distance matrix for microbial structure comparison, Bray-Curtis distance is more common.
- Figure 3 There should be more information about the meaning of these three indices in the material and methods part.
- Figure 4 I would say a total of 12-20% variation could be explained by three groups of factors is very low, any possible reasons?
- Figure 5 The exploration type is interesting to me. As the author mentioned in the introduction, they are highly correlated to the nutrient acquisition pattern of EMF, affected by both plant host and soil nutrients. But the way to visualize in a phylogenetic tree didn’t have too much information related to the study site. The author could summarize the exploration patter in individual pine forest and correlated with soil properties to understand their drivers
- Table S1 the same as Fig. 3. There should be more information about the meaning of these three indices in the material and methods part.
- Figure S2. Not very necessary.
- Figure S3. Why not using pH gradient but using 7 as a threshold? Did the author do statistics or refer to any literature to identify the pH threshold for EMF clustering? Again, it is not common to use Euclidean distance matrix for microbial structure comparison, Bray-Curtis distance is more common

Author Response
Responses to reviewers’ Comments for Adamo et al.: “Lack of phylogenetic differences in ectomycorrhizal fungi among distinct Mediterranean pine forest habitats”
Journal: Journal of Fungi Ref: jof-1308271
We would like to thank the editor and both reviewers for their valuable comments and recommendations; the inputs have helped much to improve our paper. We have carefully addressed each issue and taken the suggestions on board. We explained how we have done so. We explain this on a point by point basis.
We hope to have successfully addressed all these comments, but we would be delighted in addressing any pending issues.
We are looking forward to hearing from you.
Yours faithfully,
Irene Adamo
On behalf of all co-authors.
Dept. of Crop Production and Forest Science – AGROTECNIO Center
Universitat de Lleida (Spain)
Responses to Reviewers’ comments
Our responses are in blue.
Overall, this piece work did characterization of ectomycorrhizal fungi from five typical pine forest in Mediterranean area. This is interesting to general readers to understand their features and possible function for the plant health under this climatic condition.
We are really pleased with the reviewer’s comment stated above, we aimed to address issues that could be crucial to understand ecosystems processes in Mediterranean Pine forests.
The major concern:
- EMF are usually strong associated with plant roots. In this study, there is limited information about how their soil samples linked to their pine species host or root (seemed from top 30cm), it is difficult to make a convincing conclusion that tree host impacts were not significantly different.
We agree with the reviewer statement above, however we explained in the Material and Methods section (L: 129-135) how the soil samples were collected and how the soil physico-chemical properties were measured. Our aim was to identify whether the forest soil fungal phylogenetic communities were different, and whether pine trees have some effect over them. We were targeting the forest fungal community as a whole and not the root related communities. Moreover, in Table 1 (L. 147) we summarize the main feature of the study plots per each pine tree host. Although EMF are usually strong associated with plant roots, the study was performed in five congeneric pine forests, with similar root traits and as seen in Adamo et al. (2021). In this regard, we also think that the lack of specificity between fungal and pine species is caused by the close phylogenetic relationships among host trees (Tedersoo et al. 2013). In any case, to clarify it we have stated in the discussion (L: 400-410) that:
“with no significant differences in ectomycorrhizal phylogenetic structure and diversity between pine tree hosts. However, we observed significant differences in phylogenetic dispersion among habitats with distinct pines hosts. For example, ectomycorrhizal species in P. halepensis-nigra forest resulted phylogenetic clustered, while in P. sylvestris-nigra were slightly more over dispersed at the tip of the phylogeny (Fig.2a), probably due to the low number of P. nigra-halepensis sites which may have caused underestimation of the differences between phylogenetic taxa. In this regard, the three P. nigra-halepensis sites showed similar soil properties (i.e. values range from pH: 8.18-8.38, CN: 15-11, P: 5-2, N: 0.16-0.12), which may have resulted in the occurrence of closely related species that are adapted to these similar abiotic conditions. These results may imply that Mediterranean pine host tree species are weak habitat filters for ectomycorrhizal fungi probably due to a lack of host specificity among congeneric hosts.”
- The author did several phylogenetic and statistical analyses, but the results did not comprehensively reflect the analyses have been performed.
We thank the comment. Thus, in this new version we have tried to clarify these points.
- There are 600 abundant OTUs, which could be classified to several functional groups, but I did not see any information about relative abundance of each group.
Following the reviewer suggestion, we have now added a figure 1a displaying the relative abundance of the ectomycorrhizal family in each forest type.
- There is no detailed information about EMF profiles and characteristics from each pine forests.
We agree with the comment. Therefore, in the new manuscript version, we have added information about the most common ectomycorrhizal species in each forest type in the table S1 the supplementary information.
Table S2. Most abundant ectomycorrhizal species detected in pure and mixed stands of Pinus spp.
|
P. halepensis |
P. nigra–P. halepensis |
P. nigra |
P. sylvestris–P. nigra |
P. sylvestris |
|
Suillus spp. |
Suillus spp. |
Suillus spp. |
Suillus spp. |
Hydnum repandum |
|
Phellodon niger |
Phellodon niger |
Phellodon niger |
Phellodon niger |
Phellodon niger |
|
Rhizopogon mohelnensis |
Inocybe spp. |
Tricholoma spp. |
Tricholoma spp. |
Inocybe ochroalba |
|
Hydnum spp. |
Lactarius sanguifluus |
Hydnum repandum |
Hydnum repandum |
Tricholoma terreum |
|
Russula pallidospora |
Cortinarius hydrobivelus |
Russula spp. |
Russula pallidospora |
Russula caerulea |
|
Inocybe ochroalba |
Russula pesudoaeruginea |
Inocybe spp. |
Inocybe ochroalba |
Suillus spp. |
|
Cortinarius vernus |
Craterellus lutescens |
Craterellus cornucopioides |
Rhizopogon molhensis |
Rhizopogon molhensis |
|
Tomentella subclavigera |
Boletopsis sp. |
Boletus edulis |
Inocybe asterospora |
Russula spp. |
|
Lactarius sanguifluus |
Lactarius deliciosus |
Lactarius sanguifluus |
Sebacina flagelliformis |
Lactarius sanguifluus |
|
Craterellus lutescens |
Tricholoma sp. |
Tomentella spp. |
Tricholoma terreum |
Craterellus lutescens |
|
Russula delica |
Rhizopogon mohelnensis |
Lactarius deliciosus |
Craterellus lutescens |
Lactarius deliciosus |
|
Hydnum repandum |
Boletus edulis |
Hydnellum spp. |
Boletopsis spp. |
Sebacina spp. |
|
Lactarius deliciosus |
Cortinarius vernus |
Sebacina spp. |
Sebacina cystidiata |
Boletus edulis |
|
|
|
|
|
|
- The author measured soil texture, but where is the data? The soil moisture and texture will have major impacts on fungi due to their impacts on water and oxygen availably.
Yes, we measured soil texture, however there were no significant changes in the soil texture between forest types. Therefore, we did not include them in the analyses, however we have included this data in the supplementary.
Table S1. Table summarizing the main texture and moisture properties of the study plots: Fine clay content (%), Clay content (%), Sand content (%), Moisture (g water/g soil). Ps: P. sylvestris, Pn: P. nigra, Ph: P. halepensis, Ps-Pn: P.sylvestris-nigra, Pn-Ph: P. nigra-halepensis
|
Forest type |
Range |
Fine Lime−1 |
Clay |
Sand |
Moisture |
|
Ps |
Min. |
6.8 |
5 |
24.8 |
0.54 |
|
(14) |
Mean |
25.4 |
15.7 |
47.2 |
2.22 |
|
|
Max. |
44.9 |
31.5 |
83.4 |
4.5 |
|
Pn |
Min. |
21.5 |
3.30 |
24.7 |
1.19 |
|
(14) |
Mean |
27.95 |
14.92 |
44.04 |
2.13 |
|
|
Max. |
38.5 |
37.30 |
54.90 |
4.24 |
|
Ph |
Min. |
17.30 |
6.5 |
51.10 |
1.2 |
|
(4) |
Mean |
19.75 |
8.95 |
57 |
1.61 |
|
|
Max. |
23.20 |
11.40 |
64.80 |
2.70 |
|
Ps–Pn |
Min. |
7.50 |
9 |
25.10 |
0.78 |
|
(7) |
Mean |
21.10 |
18 |
49 |
1.80 |
|
|
Max. |
31.90 |
29.5 |
78.90 |
2.70 |
|
Pn–Ph |
Min. |
32.90 |
20.10 |
16.20 |
1.30 |
|
(3) |
Mean |
138.23 |
23.63 |
24.50 |
1.36 |
|
|
Max. |
43.70 |
29.40 |
30.90 |
1.44 |
- Some of the analyses are very confusing.
- For instance, the site description in Table 1, why using C:N ratio rather than TOC, TN, which are important to understand soil quality and fertility?
When we describe soil properties in pine stands, we use CN ratios, since these forest types are characterized usually by higher CN due to lower litter quality. Therefore, we measured CN as it is an important variable that differentiate pine stands (Awad et al. 2019). Unfortunately, we don’t have measurements of the variables suggested by the reviewer to include in our analyses.
Reference:
Awad, A., Majcherczyk, A., Schall, P., Schröter, K., Schöning, I., Schrumpf, M., and Seidel, D., 2019. Ectomycorrhizal and saprotrophic soil fungal biomass are driven by different factors and vary among broadleaf and coniferous temperate forests. Soil Biology and Biochemistry 131, 9-18. doi:10.1016/j.soilbio.2018.12.014.
- 1 phylogenetic tree, in addition to the major EMF cluster and their assigned families, it is also important to highlight whether their presence and/or relative abundance in individual forest types as it is one of the focus of this study.
Ok, following the reviewer recommendations and to make the figure clearer and more informative, we added figure 1a, in which the relative abundance of the families in each forest type is displayed.
- 2 it is not common to use Euclidean distance matrix for microbial structure comparison, Bray-Curtis distance is more common.
We agree with the reviewer that the use of Bray-Curtis distances is common in microbial community analyses. However, in this study we used a method to calculate phylogenetic community composition using an R package called philR (Silverman et al., 2017) (see Phylogenetic and statistical analyses section lines: 193-199). This package has a new multivariate method that enables to transform compositional data into an orthogonal unconstrained space with phylogenetic and evolutionary interpretation, afterwards a Redundancy analysis, (RDA function “rda”) which uses Euclidean distances, is used to visualize ectomycorrhizal phylogenetic compositional differences between tree host species.
Reference:
Silverman JD., Washburne AD, Mukherjee S, David LA 2017. A phylogenetic transform enhances analyses of compositional microbiota data. Elife. https://doi.org/10.7554/eLife.21887
- 3 There should be more information about meaning of these three indices in the material and methods part.
Thanks, we added this information in the Material and Method section, see lines 204-211:
“After calculating SES.MPD and SES.MNTD, the values were multiplied by -1 as these values are equivalent to -1 times NRI (net relatedness index) and NTI (nearest taxon index), respectively. Importantly, an increase in the NRI value indicates increasing phylogenetic clustering (or decreasing overall relatedness) of a set of species relative to the source pool [25]. On the other hand, the nearest taxon index (NTI) is a standardized measure of the mean phylogenetic distance to the nearest taxon in each sample/community [25].”
- 4 I would say a total of 12-20% variation could be explained by three groups of factors is very low, any possible reasons?
It is true. We agree that the total percentage is not high, however we argue that there may be other processes influencing ectomycorrhizal phylogenetic community, such as competition for space and resources that we were not able to test in this study. We have commented it in the discussion (L 456-459): “Finally, we hypothesize that there may be other processes influencing ectomycorrhizal phylogenetic community, such as competition for space and resources [30] that were not directly tested. Therefore, further studies are needed to further disentangle whether other processes influence phylogenetic structure in these ecosystems”.
Fig. 5 The exploration type is interesting to me. As the author mentioned in the introduction, they are highly correlated to the nutrient acquisition pattern of EMF, affected by both plant host and soil nutrients. But the way to visualize in a phylogenetic tree didn’t have too much information related to study site. The author could summarize the exploration patter in individual pine forest and correlated with soil properties to understand their drivers.
We agree that the exploration types are highly correlated with the nutrient acquisition. However, as our aim was to assess the ectomycorrhizal phylogenetic community, we described the trait evolution in order to better understand the nutrient acquisition strategy that evolved in these habitats. Therefore, to test whether there was any phylogenetic signal we used K’ Blomberg statistics to calculated for the presence of the traits. Moreover, the phylogenetic tree was used to display the traits on the tree in order to visualize the occurrence of the different traits throughout the phylogenetic tree. Unfortunately, since we found no phylogenetic signal, there was a no clear pattern in the distribution of the traits in these habitats.
- Table S1 the same as Fig. 3. There should be more information about meaning of these three indices in the material and methods part.
Thanks, we changed the explanation of the phylogenetic beta diversity indices and now it says:
L 225-228: “Moreover, to assess the phylogenetic relationships among species change across space, we computed multiple-site phylogenetic turnover, nestedness and phylo-beta diversity (Sorensen similarity index) per tree host species using “phylo.beta.multi” function in the betapart package [78].”
- S2. Not very necessary.
We added this figure (as a supplementary material) in order to show the distribution of the pH values across the five habitats as it shows that a left tail of P. sylvestris and P. sylvestris-nigra in the distribution of the pH values resulting in a wider niche for species adapted to low pH values (Fig. S2).
- S3. Why not using pH gradient but using 7 as threshold? Did the author do statistics or refer any literature to identify the pH threshold for EMF clustering? Again, it is not common to use Euclidean distance matrix for microbial structure comparison, Bray-Curtis distance is more common.
The choice to use 7 as a threshold was done because ectomycorrhizal are influenced by soil fertility gradient (Sterkenburg et al., 2015) and / is the threshold between acidic and basidic soils. However, in our study site we had pine stands where the pH was <7 and pine stands where pH was >7, regardless of the tree host. Therefore, by using this threshold we aimed to see if the phylogenetic communities were primarily clustering by pH.
References:
Sterkenburg, E., Bahr, A., Durling, M.B., Clemmensen, K.E., Lindahl, B.D., 2015. Changes in fungal communities along a boreal forest soil fertility gradient. New Phytologist. 207(4): 1145-1158. https://doi.org/10.1111/nph.13426

Reviewer 2 Report
The authors examined phylogenetic composition and structure of ectomycorrhizal fungi in Mediterranean pine forests across five vegetation types composed of three pine hosts. While no host effect was observed, environmental factor (soil pH) was suggested to have significant but weak effect on ectomycorrhizal phylogenetic structure. The authors also focused on exploration types of fungal mycelia elongated form ectomycorrhizal roots and found that short and contact types were dominated in the pine forests, suggesting their important roles in drought-prone ecosystems of Mediterranean forests.
Surely, phylogenetic community is an important aspect to understand underlying mechanisms in structuring communities and to predict future effects of habitat condition change. However, I couldn’t find the advantage of focusing on phylogenetic community not on taxonomic one in this study. In addition, discussion section should be largely reconstructed because I got the impression that obtained results were not objectively interpreted in the current manuscript, the authors’ discussion is not supported by sufficient data, and there are lots of leap of logic. In this manuscript, many typos such as misspellings and accidental deletions of space and line break were seen. Please check carefully them. I listed some comments below, so please sincerely address them.
Line-specific comments
L120 Explain the reason for high variation of the number of samples among forest types. In general, less samples would make the results obscure and decrease reliability of them.
L145 As long as I know, most related studies usually use ectomycorrhizal roots as the materials to extract the fungal DNA. Please clarify what is the strong point of directly extracting DNA from soils to survey ectomycorrhizal fungal communities.
L192 To compare community structures in RDA, Bray-Curtis dissimilarity is recommended as a community dissimilarity index especially when the community matrix has lots of zeros. Even if not in this case, B-C dissimilarity should be chosen to ensure that this result would be comparable to results of other studies.
L206 What are SES.MPD and SES.MNTD? Please spell out them in the first appearance.
P6 Use consistently the number of significant digits to indicate p-values.
Fig. 3 How the different results between NRI and NTI to be considered?
L278 Forward-selected variables in RDA should be indicated.
L294 This result of clustering according to the host trees (What is ‘tree main groups’ in the sentence?) seems to contradict NMDS result in which no significant difference among hosts in ectomycorrhizal fungal community structure. Additionally, in the sentence just below, it is too rough to divide communities into two groups at lower and higher pH values.
Fig. S3 In this figure, one community of P. halepensis belonged to the cluster of lower pH (< 7). However, the minimum value of pH of this pine forest (Ph) is indicated as 8.2 in site status of this pine forest summarized in Table 1. Please confirm it.
Fig. 5 Does this phylogenetic tree correspond to the tree of Fig. 1? To clarify the correspondence, putting family names in this tree is recommended.
L373 It seems that the family names to be indicated here is not ‘Atheliaceae…’ but ‘Bankeraceae and Thelephoraceae’, because the description about the families including Atheliaceae is found in L376.
L379 Clearly explain why the resolved phylogenetic tree could be a strong backbone for the downstream analyses. The current description is not logical, making it difficult to understand.
L393 In this study, the authors compared phylogenetic community composition of ectomycorrhizal fungi among hosts. However, the same result would be obtained if taxonomic communities were compared. It is obvious that taxonomic communities are different between hosts when difference was observed in phylogenetic communities between hosts. This means that if no different taxonomic communities were observed between hosts, there are no difference in phylogenetic communities between hosts. As referring that no differences in taxonomic communities between phylogenetically related hosts were found in the previous study, absence of phylogenetic community difference between pines in this study seems to be self-evident.
L404 What is Fig. 2a?
L412 Not logical. Please add more constructive explanations to lead this sentence.
L438 To indicate that strong effect of pH in several studies, add more references other than [86].
L455 I could not find any descriptions which support this sentence. Clarify it.
L462 As described in this paragraph, 51% of short and contact exploration types is surely high value compared with other exploration types. However, is the percentage really high value compared with other studies? Unless this point would be clarified, we cannot decide if the sentence in L476 is correct or not.
Author Response
The authors examined phylogenetic composition and structure of ectomycorrhizal fungi in Mediterranean pine forests across five vegetation types composed of three pine hosts. While no host effect was observed, environmental factor (soil pH) was suggested to have significant but weak effect on ectomycorrhizal phylogenetic structure. The authors also focused on exploration types of fungal mycelia elongated form ectomycorrhizal roots and found that short and contact types were dominated in the pine forests, suggesting their important roles in drought-prone ecosystems of Mediterranean forests.
Surely, phylogenetic community is an important aspect to understand underlying mechanisms in structuring communities and to predict future effects of habitat condition change. However, I couldn’t find the advantage of focusing on phylogenetic community not on taxonomic one in this study. In addition, discussion section should be largely reconstructed because I got the impression that obtained results were not objectively interpreted in the current manuscript, the authors’ discussion is not supported by sufficient data, and there are lots of leap of logic. In this manuscript, many typos such as misspellings and accidental deletions of space and line break were seen. Please check carefully them. I listed some comments below, so please sincerely address them.
We really appreciate the reviewer’s comments, indeed assessing phylogenetic communities is fundamental to understand ecosystems functioning and predict future effects of habitat condition change. We believed that it was important to focus this study on characterizing ectomycorrhizal phylogenetic community, with the aim of disentangling whether the environmental drivers, that influence taxonomic ectomycorrhizal communities would influence phylogenetic composition and structure. This will give us insights to better understand the mechanisms that structure ectomycorrhizal communities in Mediterranean pine forest. As far as we know, this is the first study on ectomycorrhizal phylogenetic structure and composition using ITS2 amplicons and a method as ghost-tree which sequence data to be integrated into a single tree and relating the phylogenetic community to the biotic and abiotic factor in order to understand the drivers of ectomycorrhizal communities’ structure. Therefore, we believe that the discussion was structured in order to discuss each of the results and we supported our findings with previous literature. Finally, the interpretation of the results was done based on the data provided and each section of the discussion focuses on the results we show in the results section.
Thanks for highlighting the spelling mistakes, we will carefully address these points as well as all the issue reported by the previous reviewer.
Line-specific comments
L120 Explain the reason for high variation of the number of samples among forest types. In general, less samples would make the results obscure and decrease reliability of them.
As we explained in the site selection section (L: 116-127): “The 42 pine forest were randomly selected from the 579 sites included in the Forest Ecological Inventory of Catalonia carried out by Centre de Recerca Ecològica i Aplicacions Forestals (CREAF, 1992).” The plots were randomly distributed throughout the county in numbers proportional to the area occupied by each tree species, with 14 plots corresponding to P. nigra and P. sylvestris species and 4 to P. halepensis, whilst 10 plots were mixed plots (7 mixed plots of P. sylvestris and P. nigra species and 3 plots dominated by P. nigra and P. halepensis). The preliminary idea was to collect the samples in relation of the proportional area occupied by each species to have a spatial balanced design. However, we are aware of the high variation in the number of samples; in any case, we have checked the statistical validity of our result in different ways considering the replication differences. In this regard, these results are not surprising since previous studies focusing on taxonomic communities have also reported higher levels of community compositional heterogeneity when highly diverse and distant areas are grouped together in compositional analyses (Alday et al., 2013).
Reference:
Alday, J.G., Cox, E.S., Pakeman, R.J., Harris M.P.K., Le Duc, M.G., Marrs, R.H., 2013. Overcoming resistance and resilience of an invaded community is necessary for effective restoration: a multi-site bracken control study. Journal of Applied Ecology 50, 156-167. doi:10.1111/1365-2664.12015
L145 As long as I know, most related studies usually use ectomycorrhizal roots as the materials to extract the fungal DNA. Please clarify what is the strong point of directly extracting DNA from soils to survey ectomycorrhizal fungal communities.
Being a field study, we extracted DNA directly from soil as we were interested in characterizing the overall soil fungal communities in the forest site, not only those Ectomycorrhizal species directly related with tree roots. We were applying more ecosystem community perspective rather than root communities, in any case, after identifying the soil ectomycorrhizal species, we used this data for all downstream analyses.
L192 To compare community structures in RDA, Bray-Curtis dissimilarity is recommended as a community dissimilarity index especially when the community matrix has lots of zeros. Even if not in this case, B-C dissimilarity should be chosen to ensure that this result would be comparable to results of other studies.
We agree with the reviewer that Bray-Curtis is usually used as community dissimilarity index, however our aim was to assess ectomycorrhizal community dissimilarity between habitats including the phylogenetic information. To do so, we used a novel method called philR which enables to transform compositional data into an orthogonal matrix unconstrained space with phylogenetic and evolutionary interpretation. (See Phylogenetic and Statistical analyses, L: 198-200). Also, check the above responses in this regard.
Reference:
Silverman JD., Washburne AD, Mukherjee S, David LA 2017. A phylogenetic transform enhances analyses of compositional microbiota data. Elife. https://doi.org/10.7554/eLife.21887
L206 What are SES.MPD and SES.MNTD? Please spell out them in the first appearance.
Thanks, we have added the explanation and now it reads:
L 204-208: “To test the phylogenetic structure of ectomycorrhizal communities between tree host species were calculated the standardized effect size of mean pairwise distances and mean nearest taxon distances using ses.mpd (Standardized effect of mean pairwise distances in communities) and ses.mntd (Standardized effect of nearest taxon index in communities) and ses.mntd functions from picante.”
P6 Use consistently the number of significant digits to indicate p-values.
Thanks, we have corrected it.
Fig. 3 How the different results between NRI and NTI to be considered?
NRI estimates the average phylogenetic relatedness between all possible pairs of taxa in a community, therefore at the deeper nodes of the phylogeny. The NTI calculates the mean nearest phylogenetic neighbor among the individuals in a community, therefore at the tip of the phylogeny. We have added the explanation in the Phylogenetic and statistical analyses section (L 220-223):
“The NRI measures the standardized effect size of the mean phylogenetic distance (MPD), which estimates the average phylogenetic relatedness between all possible pairs of taxa in a community. The NTI calculates the mean nearest phylogenetic neighbor among the individuals in a community.”
L278 Forward-selected variables in RDA should be indicated.
We use the RDA to identify the effect of host on fungal phylogenetic compositional differences, therefore we did not use the forward selection method. See statistical analyses section (lines 199 and 201)
L294 This result of clustering according to the host trees (What is ‘tree main groups’ in the sentence?) seems to contradict NMDS result in which no significant difference among hosts in ectomycorrhizal fungal community structure. Additionally, in the sentence just below, it is too rough to divide communities into two groups at lower and higher pH values.
Thanks, we have corrected the sentence, the phylogenetic communities clustered into clearly one group at lower pH (<7) and then a clear cluster at higher pH (>7). These results do not contradict the RDA analyses, as here pH is the only significant factor shaping the phylogenetic communities, therefore we get the two clusters (Fig S3), while Fig 2 shows that the forest type do not influence the overall phylogenetic community composition.
Fig. S3 In this figure, one community of P. halepensis belonged to the cluster of lower pH (< 7). However, the minimum value of pH of this pine forest (Ph) is indicated as 8.2 in site status of this pine forest summarized in Table 1. Please confirm it.
Thanks a lot for highlighting this, we have corrected the figure and included the right tree host (P. sylvestris) in that cluster.
Fig. 5 Does this phylogenetic tree correspond to the tree of Fig. 1? To clarify the correspondence, putting family names in this tree is recommended.
This is the phylogenetic tree at OTU level where at the tip of the phylogeny we coloured each tip with the corresponding exploration type to show the exploration type patterns throughout the phylogenetic tree. Fig 1 shows the same tree but the tips of the branches are clustered by family.
L373 It seems that the family names to be indicated here is not ‘Atheliaceae…’ but ‘Bankeraceae and Thelephoraceae’, because the description about the families including Atheliaceae is found in L376.
Thanks, we have corrected it.
L379 Clearly explain why the resolved phylogenetic tree could be a strong backbone for the downstream analyses. The current description is not logical, making it difficult to understand.
Thanks, we have clarified the importance of using a resolved phylogenetic tree in phylogenetic diversity analyses. We have changed the sentence and now it reads:
L 378-383: “Therefore, the resolved phylogenetic tree resulted in a strong backbone for the downstream analyses as the level of resolution allows to perform reliable phylogenetic diversity analyses [96]. Finally, disentangling the ectomycorrhizal phylogenetic community structure in our study region, where the current climate change may lead to changes in ecosystems functioning, is crucial to predict the impacts on ectomycorrhizal taxonomic and phylogenetic community composition and diversity [81].”
L393 In this study, the authors compared phylogenetic community composition of ectomycorrhizal fungi among hosts. However, the same result would be obtained if taxonomic communities were compared. It is obvious that taxonomic communities are different between hosts when difference was observed in phylogenetic communities between hosts. This means that if no different taxonomic communities were observed between hosts, there are no difference in phylogenetic communities between hosts. As referring that no differences in taxonomic communities between phylogenetically related hosts were found in the previous study, absence of phylogenetic community difference between pines in this study seems to be self-evident.
It is true that taxonomic and phylogenetic are in somehow related, but there are clear differences in taxonomic and phylogenetic composition. In taxonomic composition we are relating fungi in base of their similarities and dissimilarities considering only their abundance (DNA reads in our case), while in phylogenetic composition we are testing the evolutionary relationships of a group of species, which is more related with historical evolutionary processes. Thus, both could produce different results for the same dataset. Here, we aimed to test if the phylogenetic composition of ECM species was different, and if this was more related with soil rather than with host effect. The importance of our results are highlighted in the discussion section as well as the limitations in our approach. See Discussion lines: 350-365 and 487-495.
L404 What is Fig. 2a?
We have corrected it and it is fig 3a.
L412 Not logical. Please add more constructive explanations to lead this sentence.
We have changed it and now it reads:
L 505-507: “Conversely, ectomycorrhizal phylogenetic structure was marginally influenced by the combined effect of soil, stand structure and geographic distance, indicating that phylogenetic structure is mainly influenced by their combined effect.”
L438 To indicate that strong effect of pH in several studies, add more references other than [86].
Thanks, we have added other supporting literature.
References:
Lekberg, Y., Koide, R.T., Rohr, J.R., Aldrich-Wolfe, L., Morton, J.B., 2007. Role of niche restrictions and dispersal in the composition of arbuscular mycorrhizal fungal communities. Journal of Ecology 95(1), 95-105. doi:10.1111/j.1365-2745.2006.01193.x
Lladó, S., López-Mondéjar, R., Baldrian, P., 2017. Forest Soil Bacteria: Diversity, Involvement in Ecosystem Processes, and Response to Global Change. Microbiology and Molecular Biology Reviews 81(2), e0063-16. doi:10.1128/mmbr.00063-16
Kivlin SN, Winston GC, Goulden ML, Treseder KK (2014) Environmental filtering affects soil fungal community composition more than dispersal limitation at regional scales. Fungal Ecol 12:14-25. https://doi.org/10.1016/j.funeco.2014.04.004
L455 I could not find any descriptions which support this sentence. Clarify it.
Ok, we have included a reference to validate it.
Pescador David S., de Bello Francesco, López-Angulo Jesús, Valladares Fernando, Escudero Adrián (2021) Spatial Scale Dependence of Ecological Factors That Regulate Functional and Phylogenetic Assembly in a Mediterranean High Mountain Grassland. Frontiers in Ecology and Evolution, 9: 482. DOI=10.3389/fevo.2021.622148
L462 As described in this paragraph, 51% of short and contact exploration types is surely high value compared with other exploration types. However, is the percentage really high value compared with other studies? Unless this point would be clarified, we cannot decide if the sentence in L476 is correct or not.
Previous studies (Castaño et al 2018), showed that in Mediterranean pine forests short exploration types are dominant exploration types. However, as we explained, our results showed that traits were dispersed across the phylogenetic tree, therefore, this trait dispersion across the phylogenetic tree suggests that even more distant related species showed the same exploration type, resulting in a random trait pattern with lack of phylogenetic signal. Therefore, even though being the most abundant trait, we detected no clear evolution pattern of this trait in these habitats. In contrast, our results suggest that the abundance of short and contact exploration types (short-contact types) may be related with drier conditions in these forest ecosystems.
References:
Castaño C, Lindahl BD, Alday JG, et al (2018) Soil microclimate changes affect soil fungal communities in a Mediterranean pine forest. New Phytol 220(4):1211-1221. https://doi.org/10.1111/nph.15205

Round 2
Reviewer 2 Report
Most of the issues found in the previous review have been addressed and nicely improved, but there are still some problems remained to be modified.
L324 Insert a space between number and unequal sign (‘>’ or ‘<’). The same modifications are needed also in other lines.
L278 Forward-selected variables in RDA should be indicated.
We use the RDA to identify the effect of host on fungal phylogenetic compositional differences, therefore we did not use the forward selection method. See statistical analyses section (lines 199 and 201)
The sentence of line 244–245 says that some explanatory variables were forward-selected for variation partitioning analysis. Please clarify this discrepancy.
Author Response
Responses to reviewers’ Comments for Adamo et al.: “Lack of phylogenetic differences in ectomycorrhizal fungi among distinct Mediterranean pine forest habitats”
Journal: Journal of Fungi Ref: jof-1308271
Most of the issues found in the previous review have been addressed and nicely improved, but there are still some problems remained to be modified.
Thanks, we have carefully addressed all the issue and we are glad to solve the problems that still remain in order to really improve the final manuscript.
L324 Insert a space between number and unequal sign (‘>’ or ‘<’). The same modifications are needed also in other lines.
Yes, we have changed all the lines needed to be changed.
L278 Forward-selected variables in RDA should be indicated.
We use the RDA to identify the effect of host on fungal phylogenetic compositional differences, therefore we did not use the forward selection method. See statistical analyses section (lines 199 and 201)
The sentence of line 244–245 says that some explanatory variables were forward-selected for variation partitioning analysis. Please clarify this discrepancy.
Yes, we explain in line 235-240: “The geographical distances included were previously evaluated using principal coordinates of neighbours’ matrices spatial eigenvectors (PCNM, pcnm function) based on UTM coordinates of the sampled stands with Euclidean distances. Thus, significant spatial eigenvectors were forward selected to be used as explanatory variables in the variation partitioning”. Therefore, the forward selection was done only on the spatial eigenvectors to keep only the significant ones. After keeping the significant ones, we ran the variation partitioning with the soil and stand structure variables, these variables were not forward selected. To clarify this point, we changed the sentence and now it reads:
L 240-244: “Thus, significant spatial eigenvectors were forward selected and the selected spatial eigenvectors were used as explanatory variables in the variation partitioning, together with soil (Sand content, K, Mg, organic matter, Na, N, P, water pH and CN ratio) and stand structural variables (Tree species, Altitude, Slope, Trees per hectare and Basal Area).”
